# Physical activity in adult users of inpatient mental health services: A scoping review

**Garry A. Tew[1]\*, Emily Peckham[2], Suzy Ker[3], Jo Smith[3], Philip Hodgson[1,3], Katarzyna K. Machaczek[4], Matthew Faires[1]**

1 Institute for Health and Care Improvement, York St John University, York, United Kingdom, 2 School of Medical and Health Sciences, Bangor University, Bangor, United Kingdom, 3 Tees Esk and Wear Valleys NHS Foundation Trust, Darlington, United Kingdom, 4 College of Health, Wellbeing and Life Sciences, Sheffield Hallam University, Sheffield, United Kingdom

\* g.tew@yorksj.ac.uk

**Data Availability Statement:** All relevant data are within the paper and its Supporting information files.

## Abstract

People with severe mental illness engage in considerably less physical activity than those without. They also experience premature mortality of around 10–25 years. A large proportion of these premature deaths are attributed to modifiable behaviours, including physical activity. The inpatient environment provides an opportunity to support people to become more physically active; however, there is limited evidence on which interventions are most successful and what contextual factors affect their delivery. A scoping review was conducted to help understand the extent and type of evidence in this area and identify research gaps. We included studies of physical activity correlates and interventions in adult inpatient mental health services published in peer-reviewed journals. Reviews, meta-analyses, and papers focusing on eating disorder populations were excluded. We searched the MEDLINE, CINAHL, PsycINFO, ASSIA and Web of Science databases for relevant studies published in English. We extracted data on study design, participant characteristics, intervention and control conditions, key findings, and research recommendations. We used a descriptive analytical approach and results are presented in tables and figures. Of 27,286 unique records screened, 210 reports from 182 studies were included. Sixty-one studies reported on correlates of physical activity, and 139 studies reported on physical activity interventions. Most intervention studies used a single-group, pre-post design (40%) and included fewer than 100 participants (86%). Ninety percent of interventions delivered physical activity directly to participants, and 50% included group-based sessions. The duration, type, frequency and intensity of sessions varied. Mental health was the most commonly reported outcome (64%), whereas physical activity was rarely an outcome (13%). Overall, there is a modest but growing body of research on physical activity in adult users of inpatient mental health services. More high-quality trials are needed to advance the field, and future research should target neglected intervention types, outcomes, populations and settings.

**Funding:** This study was supported by Research Capability Funding from Tees Esk and Wear Valleys NHS Trust The funders had no role in study design, data collection and analysis, decision to publish, or preparation of the manuscript.

**Competing interests:** The authors have declared that no competing interests exist.

## Introduction

People who use mental health services, including people with severe mental illness (SMI), depression, anxiety or stress disorders and people with alcohol and substance use disorders, experience worse health outcomes and a shortened life expectancy when compared to people without these disorders (10–25 years for SMI, 7–20 years for anxiety/stress disorders and alcohol and substance use disorders) [1–4]. Much of this reduced life expectancy is due to modifiable risk factors such as engaging in health risk behaviours, one of which is physical inactivity [5–7]; and it has been reported that between 70 and 75% of people with schizophrenia do not meet public health targets for physical activity [8]. Yet, physical activity interventions have been shown to improve cardiometabolic health outcomes in people with mental illnesses and improve symptoms of depression, cognitive function, feelings of isolation, and quality of life [9–11].

Insufficient physical activity and excessive sedentary behaviour in people receiving care for SMI have been observed in both community and inpatient settings [12,13]. However, the majority of intervention studies have recruited people receiving care in community and outpatient settings, and less research has been done in inpatient settings [14]. While some of the reasons for insufficient physical activity and excessive sedentary behaviour are common across both inpatient and community settings, such as the sedative effects of psychotropic medicine [15] and mental health symptoms (lack of motivation, anhedonia, fatigue and negative body image) [13], there are some unique barriers presented by inpatient settings. For example, lack of appropriate facilities, lack of suitably trained staff to support physical activity, and restrictions on leave for people detained under mental health legislation all contribute to decreased opportunities for physical activity in inpatient settings [16–19]. However, inpatient physical activity programmes provide an opportunity to improve patients' physical and mental health. For this reason, it is important to have an up-to-date understanding of the literature on this topic.

The scope of inpatient physical activity research is broad, including quantitative or qualitative research such as randomised controlled trials (RCTs) of physical activity interventions or interviews with service users exploring their perceptions of physical activity. Studies have also explored correlates of physical activity or been more intervention focused. Qualitative studies have examined the views of patients, carers or healthcare professionals, or a particular subpopulation of patients. Recent reviews regarding inpatient physical activity have focused on tightly defined inclusion criteria leading to the inclusion of specific sets of studies, such as RCTs. One systematic review and meta-analysis [14] explored the benefits of, adherence to, and safety of physical activity interventions delivered in inpatient mental health settings. This review and meta-analysis also investigated trials that supported sustaining physical activity after patient discharge and discussed patient feedback on physical activity interventions [14]. In addition, two reviews focused on physical activity interventions for inpatients in secure forensic settings. The first of these reviews investigated the effectiveness of physical activity programmes for inpatients in secure forensic settings on various health outcomes [20]. The second paper used the scoping review methodology to explore and synthesise the literature on physical activity interventions for inpatients in secure mental health settings [21]. [14,20,21] A broader review of inpatient physical activity research, that considers all inpatient settings and quantitative and qualitative evidence would be advantageous to understand the current state of the literature and inform future research. A scoping review is a suitable way of achieving this [22]. Understanding and mapping the available evidence on physical activity in inpatient settings, including qualitative and quantitative research, is important due to their complementary nature. They answer different questions about physical activity. For example, quantitative research can

shed light on cause-and-effect relationships between various factors associated with physical activity, while qualitative research can tell us why this is the case. A scoping review also provides a means to determine whether there is scope and a need for a systematic review of a particular type or in a specific area of literature, and to identify gaps in the existing research base that could be filled by future primary research [22]. The overall aim of this scoping review was therefore to understand the extent and type of evidence regarding physical activity in adult users of inpatient mental health services. This included study designs used, primary conditions of participants, types of outcomes assessed, correlates of physical activity explored, intervention characteristics, outcomes, and research recommendations.

## Methods

This scoping review was conducted in accordance with the Joanna Briggs Institute (JBI) methodology for scoping reviews [23] and reported in accordance with the PRISMA Extension for Scoping Reviews (PRISMAScR) [24]. A protocol was prepared in advance and published in the Open Science Framework [25].

### Participants/Context

We included studies pertaining to adult ($\geq$18 years) users of inpatient mental health services. This included studies where psychiatric inpatients were the participants and studies where other stakeholders were involved (e.g., healthcare professionals giving their views regarding services for psychiatric inpatients). Studies focusing on learning disability populations were included whereas those focusing on eating disorder populations were excluded because of the unique requirement to carefully manage energy input/output in these populations. "Inpatient Setting" was defined as mental health care facilities which provide continuous care for a period of over 24 hours. This included psychiatric hospitals, separate inpatient units of a general hospital, residential treatment centres, and the prison service. Outpatient and community living participant-based studies were not included. There were no limits on the country of origin.

### Concept

The phenomenon of interest was physical activity. Physical activity has been defined as any bodily movement produced by skeletal muscles and requiring energy expenditure [26]. Exercise is a subset of physical activity that has been defined as any structured and repetitive physical activity that has an objective of improving or maintaining physical fitness [26]. To address the aims of the review we took a broad view of physical activity and included studies that had focused on supervised exercise or promotion of self-managed physical activity.

### Types of sources

This scoping review included experimental and quasi-experimental study designs such as RCTs, non-randomised controlled trials, before and after studies and interrupted time-series studies. In addition, analytical observational studies including prospective and retrospective cohort studies, case-control studies and analytical cross-sectional studies were included. This review also considered descriptive observational study designs including case series, individual case reports and descriptive cross-sectional studies for inclusion. Qualitative studies that utilised various methodologies (e.g., phenomenology, grounded theory, ethnography, action research) were also considered. Literature reviews and meta-analyses were used to identify primary studies, but were excluded from data analysis. Conference abstracts and opinion papers were also excluded.

## Search strategy

The search strategy targeted peer-reviewed publications. A pilot search of the MEDLINE database was undertaken to identify articles on the topic. The text words contained in the titles and abstracts of relevant articles, and the index terms used to describe them were employed to develop a full search strategy for the MEDLINE, CINAHL, PsycINFO, ASSIA and Web of Science databases (S1 File). The search strategy, including all identified keywords and index terms, was adapted for each included database and/or information source. The database searches were conducted on October 31, 2022 and updated on October 24, 2023. The reference list of all included sources of evidence, and those of any review articles or meta-analyses, were screened for additional studies. We also conducted forward citation tracking of included studies using Google Scholar. We only included studies that were published in English language from 2007 onwards. The latter relates to amendments in the 1983 Mental Health Act that were made in 2007; prior studies may have employed different clinical practices and patient populations.

## Study selection

Following the search, all identified citations were collated and uploaded into Covidence [27], and duplicates were removed. After a pilot test, the titles and abstracts were screened by pairs of independent reviewers against the eligibility criteria for the review. Potentially relevant sources were retrieved in full and assessed in detail against the eligibility criteria by pairs of independent reviewers. The reasons for excluding sources of evidence in full text that did not meet the eligibility criteria were documented. Any disagreements that arose between the reviewers at each stage of the selection process were resolved through discussion or with the aid of additional reviewers.

## Data extraction

Data were extracted from papers included in the scoping review by pairs of independent reviewers using a data extraction form (S2 File) developed by the reviewers in Covidence [27], which was based on the JBI template extraction tool [28]. The form was piloted on twenty papers to ensure it was fit for purpose.

The extracted data encompassed specific details about the participants, concept, context, study methods, key findings relevant to the review questions, and research recommendations. Any disagreements between the reviewers were resolved through discussion or with the aid of an additional reviewer. Due to limited timescales and resources, we did not contact any authors to request missing or additional data.

## Data analysis and presentation

A descriptive analytical approach was used to summarise the included studies' contextual, process and outcome-related data [29,30]. This approach was undertaken to map the key concepts and available evidence, synthesise existing research findings, and identify research gaps. Extracted data were organised in Microsoft Excel. Physical activity correlates were categorised based on the Socio-Ecological Model [31–33]. This model was chosen as it considers the multifaceted and interactive effects of various factors (e.g., interpersonal, organisational and community), which characterise the delivery of physical activity interventions for people with SMI. Themes for research recommendations were derived inductively following review of original quotes by two independent reviewers. The analysis was reported in accordance with the synthesis without meta-analysis (SWiM) guideline [34], with data presented in tables and figures where appropriate.

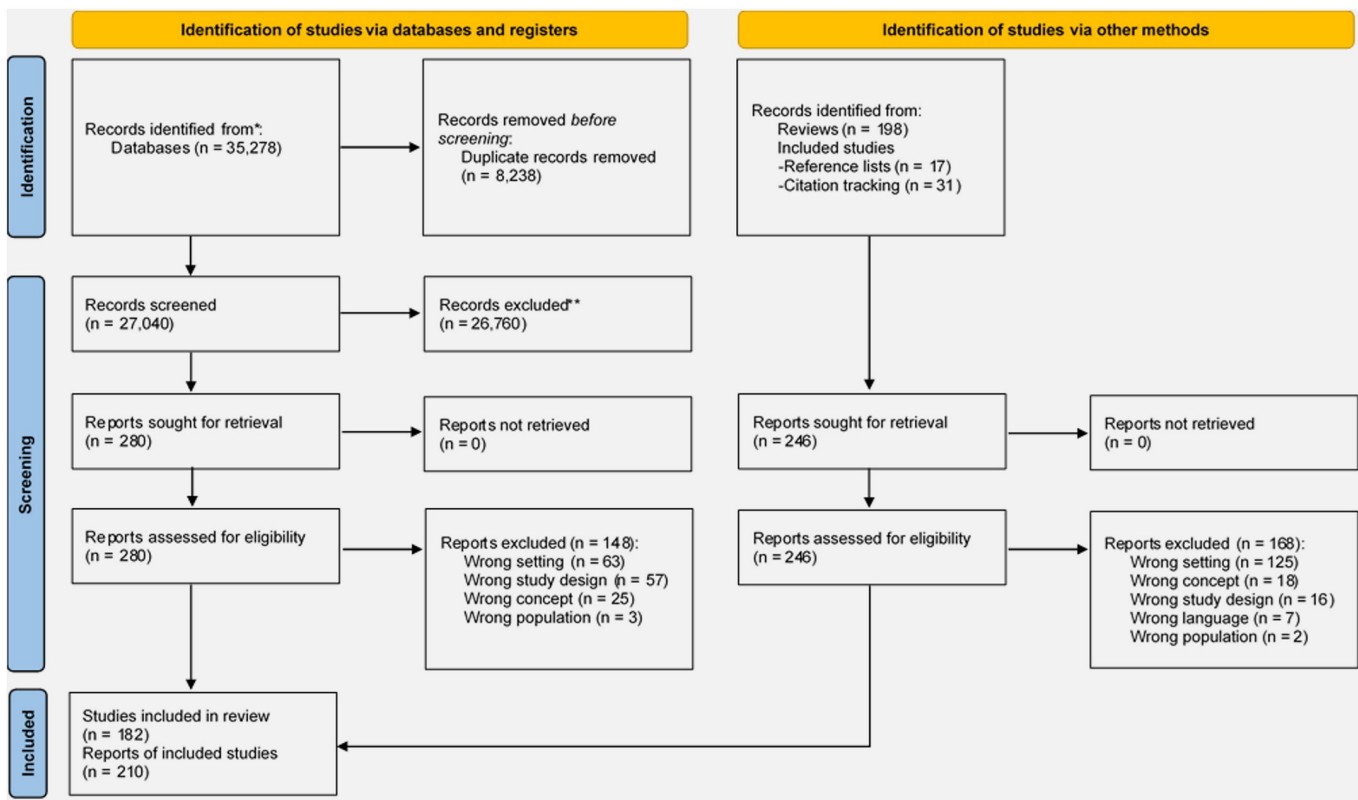

**Fig 1. PRISMA flow diagram.**

## Results

Fig 1 shows the PRISMA flow diagram. The database searches yielded 35,278 records. After removing duplicates, we screened 27,040 records, from which we reviewed 280 full-text reports, and finally included 132 reports. Later, we reviewed a further 246 full-text reports that were identified from review articles or the forward and backward citations of included reports. This resulted in a further 78 reports being included. Together, 210 reports from 182 studies were included in the review. The complete reference list of included reports can be found in S3 File.

### Descriptive characteristics and methodological information of included articles

Sixty-one (34%) studies reported on correlates of physical activity, 139 (76%) studies reported on physical activity interventions, and 19 (10%) studies reported on both physical activity correlates and interventions.

Of the 61 studies on correlates of physical activity, 43 (70%) studies reported on quantitative data (28 observational studies, 15 interventional studies) and 22 (36%) studies reported on qualitative data (14 observational studies, 8 interventional studies) (Table 1). Sample sizes varied by study type, with observational quantitative studies including the largest samples. The majority of studies were conducted in Europe (62%), with most of the remainder originating from Australasia (16%) and Asia (11%). The three most common settings were psychiatric hospitals (39%), forensic/secure settings (23%), and 'mixed' settings (16%).

**Table 1. Characteristics of studies of physical activity correlates by type of study\*.**

| | Quantitative | | Qualitative | | Overall (n = 61) |
|---|---|---|---|---|---|
| | Observational (n = 28) | Interventional (n = 15) | Observational (n = 14) | Interventional (n = 8) | |
| Sample size | 1–100: 11 (39%) | 1–100: 12 (80%) | 1–15: 6 (43%) | 1–15: 3 (37.5%) | 1–100: 40 (65.5%) |
| | 101–500: 15 (54%) | 101–500: 3 (20%) | 16+: 8 (57%) | 16+: 4 (50%) | 101–500: 18 (29.5%) |
| | 500+: 2 (7%) | 500+: 0 (0%) | Missing: 0 (0%) | Missing: 1 (12.5%) | 500+: 2 (3%) |
| | | | | | Missing: 1 (2%) |
| Continent | | | | | |
| Europe | 16 (57%) | 12 (80%) | 7 (50%) | 6 (75%) | 38 (62%) |
| North America | 1 (4%) | 2 (13%) | 0 (0%) | 0 (0%) | 3 (5%) |
| South America | 0 (0%) | 0 (0%) | 0 (0%) | 0 (0%) | 0 (0%) |
| Asia | 4 (14%) | 0 (0%) | 2 (14%) | 1 (12.5%) | 7 (11%) |
| Australasia | 5 (18%) | 1 (7%) | 4 (29%) | 1 (12.5%) | 10 (16%) |
| Africa | 1 (4%) | 0 (0%) | 1 (7%) | 0 (0%) | 2 (3%) |
| Multiple continents | 1 (4%) | 0 (0%) | 0 (0%) | 0 (0%) | 1 (2%) |
| Setting | | | | | |
| Psychiatric hospital | 13 (46%) | 8 (53%) | 3 (21%) | 3 (37.5%) | 24 (39%) |
| General hospital | 1 (4%) | 2 (13%) | 1 (7%) | 2 (25%) | 6 (10%) |
| Mental health rehabilitation centre | 3 (11%) | 1 (7%) | 0 (0%) | 0 (0%) | 4 (7%) |
| Forensic/secure | 3 (11%) | 3 (20%) | 5 (36%) | 3 (37.5%) | 14 (23%) |
| Mixed | 5 (18%) | 1 (7%) | 5 (36%) | 0 (0%) | 10 (16%) |
| Other | 2 (7%) | 0 (0%) | 0 (0%) | 0 (0%) | 2 (3%) |
| Missing | 1 (4%) | 0 (0%) | 0 (0%) | 0 (0%) | 1 (2%) |

\*Numbers are not exclusive–studies can be counted multiple times, for example, if they include both quantitative and qualitative data.

Of the 139 studies on interventions (Table 2), 57 (41%) studies used a single-group, pre-post design and 57 (41%) studies were RCTs. The majority of studies included fewer than 100 participants (86%) and were conducted in Europe (47%) and within a psychiatric hospital (45%). The most common study populations included people with schizophrenia or related psychotic disorders (35%) and a mixture of diagnoses (20%). Mental health was the most commonly reported outcome (64%), whereas physical activity (13%) and quality of life (16%) were rarely reported outcomes (Table 2).

Fourteen (10%) of the intervention studies were qualitative studies. All 14 studies collected qualitative data from service users who had participated in the intervention; seven studies (50%) also collected data from healthcare staff, and one study (7%) also collected data from family members. Most studies (71%) used one-to-one interviews to collect data. Other methods included observations (n = 2), evaluation forms (n = 2), document analysis (n = 1) and focus groups (n = 1).

## Correlates of physical activity

Tables 3 and 4 summarise the physical activity correlates examined or highlighted in quantitative and qualitative studies, respectively. A broad range of individual, interpersonal, environmental and organisational correlates was reported regarding inpatient physical activity, whereas no wider societal factors were highlighted. 'Health status' and 'medication side effects' were the most commonly reported demographic and biological factors in both quantitative (23% and 21%, respectively) and qualitative (55% and 23%, respectively) studies. 'Self-motivation' was the most common factor in the 'psychological, cognitive and emotional' category

**Table 2. Characteristics of studies of physical activity interventions by type of study\*.**

| | Randomised controlled trials (n = 57) | Other design (n = 84) | Overall (n = 139) |
|---|---|---|---|
| Sample size | 1–100: 49 (86%) | 1–100: 71 (84.5%) | 1–100: 119 (86%) |
| | 101–500: 8 (14%) | 101–500: 10 (12%) | 101–500: 17 (12%) |
| | | Missing: 3 (3.5%) | Missing: 3 (2%) |
| **Continent** | | | |
| Europe | 24 (42%) | 44 (52%) | 66 (47%) |
| North America | 4 (7%) | 13 (15%) | 17 (12%) |
| South America | 3 (5%) | 1 (1%) | 4 (3%) |
| Asia | 24 (42%) | 13 (15%) | 37 (27%) |
| Australasia | 2 (4%) | 12 (14%) | 14 (10%) |
| Africa | 0 (0%) | 1 (1%) | 1 (1%) |
| Multiple continents | 0 (0%) | 0 (0%) | 0 (0%) |
| **Setting** | | | |
| Psychiatric hospital | 28 (49%) | 35 (42%) | 63 (45%) |
| General hospital | 16 (28%) | 18 (21%) | 34 (24%) |
| Forensic/secure | 1 (1%) | 15 (18%) | 16 (12%) |
| Mental health rehabilitation centre | 9 (16%) | 13 (15%) | 22 (16%) |
| Mixed | 1 (1%) | 2 (2%) | 3 (2%) |
| Other | 2 (4%) | 1 (1%) | 3 (2%) |
| **Primary condition** | | | |
| Depression or bipolar disorder | 17 (30%) | 10 (12%) | 27 (19%) |
| Schizophrenia or related psychotic disorders | 23 (40%) | 25 (30%) | 48 (35%) |
| Post-Traumatic Stress Disorder | 1 (2%) | 1 (1%) | 2 (1%) |
| Dementia | 2 (4%) | 1 (1%) | 3 (2%) |
| Substance misuse | 4 (7%) | 3 (4%) | 7 (5%) |
| Personality disorder | 0 (0%) | 1 (1%) | 1 (1%) |
| Learning disability | 0 (0%) | 1 (1%) | 1 (1%) |
| Mixed | 6 (11%) | 22 (26%) | 28 (20%) |
| Unclear | 4 (7%) | 20 (24%) | 24 (17%) |
| **Type of programme** | | | |
| Physical activity delivered | 46 (81%) | 67 (80%) | 113 (81%) |
| Physical activity promoted | 7 (12%) | 4 (5%) | 11 (8%) |
| Mixed | 3 (5%) | 10 (12%) | 13 (9%) |
| Unclear | 1 (2%) | 3 (4%) | 4 (3%) |
| **Type of physical activity** | | | |
| Various activities (including the other categories listed) | 11 (19%) | 22 (26%) | 33 (24%) |
| Structured exercise | 22 (39%) | 26 (31%) | 48 (34.5%) |
| Exer-gaming | 1 (2%) | 4 (5%) | 5 (4%) |
| Sport | 2 (3.5%) | 8 (9.5%) | 10 (7%) |
| Yoga | 3 (5%) | 8 (9.5%) | 11 (8%) |
| Unstructured walking/ jogging/ running | 12 (21%) | 5 (6%) | 17 (12%) |
| Dance | 4 (7%) | 6 (7%) | 8 (6%) |
| Unclear | 2 (3.5%) | 5 (6%) | 7 (5%) |
| **Duration of programme** | | | |
| Variable | 0 (0%) | 7 (8%) | 7 (5%) |
| Duration of admission | 2 (4%) | 17 (20%) | 19 (14%) |
| Fixed (≤6 weeks) | 24 (42%) | 8 (9.5%) | 32 (23%) |
| Fixed (>6 weeks) | 26 (46%) | 32 (38%) | 58 (42%) |

(*Continued*)

**Table 2.** (Continued)

| | Randomised controlled trials (n = 57) | Other design (n = 84) | Overall (n = 139) |
|---|---|---|---|
| Single session | 5 (9%) | 11 (19%) | 16 (11.5%) |
| Unclear | 0 (0%) | 9 (11%) | 9 (6.5%) |
| Frequency of sessions | | | |
| Single session | 4 (7%) | 10 (12%) | 14 (10%) |
| Two sessions | 1 (2%) | 0 (0%) | 1 (1%) |
| Three sessions | 1 (2%) | 0 (0%) | 1 (1%) |
| 1 x week | 1 (2%) | 7 (8%) | 8 (6%) |
| 2 x week | 10 (17.5%) | 13 (15.5%) | 23 (16.5%) |
| 3 x week | 19 (33%) | 15 (18%) | 34 (24.5%) |
| 4 x week | 2 (3.5%) | 1 (1%) | 3 (2%) |
| 5 x week | 5 (9%) | 6 (7%) | 11 (8%) |
| 6 x week | 0 (0%) | 1 (1%) | 1 (1%) |
| 7 x week | 7 (12%) | 6 (7%) | 13 (9%) |
| Mixed frequency | 0 (0%) | 8 (9.5%) | 8 (6%) |
| Unclear | 7 (12%) | 17 (20%) | 24 (17%) |
| Duration of sessions | | | |
| 0–30 mins | 17 (30%) | 12 (14%) | 29 (21%) |
| 31–60 mins | 28 (49%) | 44 (52%) | 72 (52%) |
| 60+ mins | 2 (3.5%) | 4 (5%) | 6 (4%) |
| Variable | 1 (2%) | 2 (2%) | 3 (2%) |
| Unclear | 9 (16%) | 22 (26%) | 31 (22%) |
| Intensity of physical activity | | | |
| Self-selected | 4 (7%) | 7 (8%) | 11 (8%) |
| Prescribed | 34 (60%) | 20 (24%) | 54 (39%) |
| Unclear | 19 (33%) | 57 (68%) | 76 (55%) |
| Provider | | | |
| Study team | 15 (26%) | 6 (7%) | 21 (15%) |
| Qualified exercise therapist (internal) | 16 (28%) | 27 (32%) | 42 (30%) |
| Qualified exercise therapist (external) | 3 (5%) | 10 (12%) | 12 (9%) |
| Physiotherapist | 1 (2%) | 6 (7%) | 7 (5%) |
| Occupational therapist | 2 (3.5%) | 7 (8%) | 9 (6.5%) |
| Mental health worker | 6 (10.5%) | 7 (8%) | 13 (9%) |
| Mental health worker trained by study team | 3 (5%) | 4 (5%) | 7 (5%) |
| Unclear | 11 (19%) | 17 (20%) | 28 (20%) |
| Who with? | | | |
| Individual | 7 (12%) | 10 (12%) | 17 (12%) |
| Group | 24 (42%) | 45 (54%) | 69 (50%) |
| Mixed | 4 (7%) | 9 (11%) | 13 (9%) |
| Unclear | 22 (39%) | 20 (24%) | 42 (30%) |
| Comparator | | | |
| No comparator | 0 (0%) | 55 (65%) | 55 (40%) |
| Usual care | 25 (44%) | 16 (19%) | 41 (29%) |
| Other PA intervention | 11 (19%) | 4 (5%) | 15 (11%) |
| Other non-PA intervention | 11 (19%) | 6 (7%) | 17 (12%) |
| Other PA and usual care | 6 (11%) | 2 (2%) | 8 (6%) |
| Other non-PA and usual care | 2 (4%) | 0 (0%) | 2 (1%) |
| Other PA and other non-PA intervention | 1 (2%) | 0 (0%) | 1 (1%) |

(*Continued*)

**Table 2.** (Continued)

|  | Randomised controlled trials (n = 57) | Other design (n = 84) | Overall (n = 139) |
|---|---|---|---|
| Unclear | 1 (2%) | 1 (1%) | 2 (1%) |
| Outcomes |  |  |  |
| PA self-report | 4 (7%) | 7 (8%) | 11 (8%) |
| PA device-based | 2 (3.5%) | 5 (6%) | 7 (5%) |
| PA mixed | 2 (3.5%) | 2 (2%) | 4 (3%) |
| Mental health | 40 (70%) | 49 (58%) | 89 (64%) |
| Physical health | 22 (39%) | 29 (35%) | 51 (37%) |
| Cognition | 11 (19%) | 1 (1%) | 12 (9%) |
| Quality of life | 11 (19%) | 11 (13%) | 22 (16%) |

*Numbers are not exclusive–studies can be counted multiple times.

PA, Physical activity.

'PA promoted' refers to interventions such as health education, coaching and physical activity referral designed to increase total levels of physical activity. 'PA delivered' refers to interventions, such as structured exercise like strength, balance, functional and/or resistance training, where the activity is provided directly to the participant.

(37% quantitative, 45% qualitative). Factors were less frequently reported in the 'behavioural' and 'social and cultural' categories, but 'physical activity enjoyment' (7% quantitative, 5% qualitative) and 'social support' (26% quantitative, 41% qualitative) were the most common factors, respectively. 'Access to equipment and facilities' (28% quantitative) and 'environment restrictions' (23% quantitative, 32% qualitative) were commonly cited physical environment factors. 'Staff capacity' (33% quantitative, 82% qualitative) and 'staff capability' (51% quantitative, 59% qualitative) were among the top-ranking organisational factors.

Table 5 summarises the themes of research recommendations from the studies of physical activity correlates. Twenty-two (36%) of the 61 studies included research recommendations, which we grouped according to whether they were related to the research topic, research methods, or patient involvement.

## Physical activity interventions

The components of physical activity interventions are summarised in Table 2. Most interventions delivered physical activity directly to participants (90%) and in group-based sessions (50%). Most physical activity interventions included structured exercise (34.5%) such as aerobic and/or resistance training, or a variety of physical activities (24%). Fewer studies considered sport- or dance-based interventions (7% and 6%, respectively). The duration, frequency and intensity of sessions varied (see Table 2). Most interventions were delivered by health and/or exercise professionals, but few studies assessed interventions *for* staff. There was also very little evidence on environmental interventions with only one study exploring an environmental intervention.

Table 6 summarises the themes of the findings from the 14 qualitative studies, which we grouped under the headings of 'perceptions of the intervention' and 'factors influencing intervention delivery and participation'. Beneficial psychosocial effects of interventions were reported in most studies (79%), such as participants feeling more relaxed, happier and calmer. Several studies reported on a range of personal and environmental factors that might influence participation in physical activity. Examples of personal factors included perceived health benefits, social support, medication side effects, confinement in a locked facility, and the availability of trained staff. Environmental factors included physical space and facilities.

**Table 3. Summary of physical activity correlates examined or highlighted in quantitative studies.**

| Factor | Number (%) of studies out of 43 |
|---|---|
| Demographic and biological factors | |
| Health status | 10 (23%) |
| Medication side-effects | 9 (21%) |
| Weight/fatness | 5 (12%) |
| Financial status | 4 (9%) |
| Age | 4 (9%) |
| Gender | 4 (9%) |
| Physical activity status of healthcare staff | 3 (7%) |
| Education | 3 (7%) |
| Insomnia | 2 (5%) |
| Fitness status | 1 (2%) |
| Diabetes | 1 (2%) |
| Marital status | 1 (2%) |
| Anti-psychotic medication dose | 1 (2%) |
| Legal guardianship status | 1 (2%) |
| Residential status | 1 (2%) |
| Duration of treatment | 1 (2%) |
| Psychological, cognitive and emotional factors | |
| Self-motivation | 16 (37%) |
| Psychological health | 7 (16%) |
| Self-efficacy | 7 (16%) |
| Attitudes towards physical activity | 6 (14%) |
| Outcome expectations | 6 (14%) |
| Time availability | 5 (12%) |
| Mental illness stigma | 5 (12%) |
| Mood | 5 (12%) |
| Physical activity knowledge | 3 (7%) |
| Fear of injury | 3 (7%) |
| Fatigue/Tiredness | 3 (7%) |
| Value of physical health benefits | 2 (5%) |
| Cognitive function | 1 (2%) |
| Fear of going outdoors | 1 (2%) |
| Attitudes to staff support | 1 (2%) |
| Self-consciousness | 1 (2%) |
| Mental health condition | 1 (2%) |
| Severity of dependence | 1 (2%) |
| Behavioural factors | |
| Physical activity enjoyment | 3 (7%) |
| Alcohol use | 3 (7%) |
| Drug use | 2 (5%) |
| Physical activity duration | 1 (2%) |
| Physical activity intensity | 1 (2%) |
| Physical activity autonomy | 1 (2%) |
| Sleep | 1 (2%) |
| Smoking habits | 1 (2%) |
| Dietary habits | 1 (2%) |
| Social and cultural factors | |
| Social support from family, friends, or peers | 11 (26%) |

(*Continued*)

**Table 3.** (Continued)

| Factor | Number (%) of studies out of 43 |
|---|---|
| Staff influence/involvement | 5 (12%) |
| Someone to exercise with | 1 (2%) |
| Acceptance of women exercising | 1 (2%) |
| Acceptance of group exercise | 1 (2%) |
| Religious beliefs | 1 (2%) |
| Physical environment factors | |
| Access to equipment and facilities | 12 (28%) |
| Environment restrictions | 10 (23%) |
| Safe environment | 4 (9%) |
| Unhealthy environment | 2 (5%) |
| Weather/climate | 1 (2%) |
| Access to green space | 1 (2%) |
| Organisational factors | |
| Staff capability to support physical activity | 22 (51%) |
| Staff availability/capacity | 14 (33%) |
| Attitudes to/priority for physical activity | 10 (23%) |
| Staff responsibility for physical activity | 6 (14%) |
| Funding | 6 (14%) |
| Opportunities for physical activity | 5 (12%) |
| Staff confidence to support physical activity | 2 (5%) |
| Staff working collaboratively | 1 (2%) |
| Staff awareness of physical health issues | 1 (2%) |
| Hospital size | 1 (2%) |
| Safeguarding procedures | 1 (2%) |
| Awareness of opportunities | 1 (2%) |
| Timing of physical activity sessions | 1 (2%) |
| Length of inpatient stay | 1 (2%) |

Table 7 summarises the themes of research recommendations from the studies of physical activity interventions. Forty-nine (35%) of the 139 studies included research recommendations, which we grouped according to whether they were related to the research topic or research methods. The most commonly reported research topic-related recommendation was to investigate other physical activity interventions (11.5%), such as different modes of exercise or different behaviour change strategies. The most commonly reported research methods-related recommendations were longer follow-up periods (12%), investigating other outcomes (11.5%), and larger sample sizes (11%).

## Discussion

### Summary of evidence

This scoping review shows that a wide range of research has been published regarding physical activity and adult users of inpatient mental health services since 2007.

Forty-three quantitative studies explored correlates of physical activity. Organisational factors and psychological, cognitive and emotional factors were the top two domains cited (Table 3). Within these, four out of the top five factors were organisational factors: staff capability, staff capacity, staff attitudes to physical activity, and access to equipment, while self-

**Table 4. Summary of physical activity correlates examined or highlighted in qualitative studies.**

| Factor | Number (%) of studies out of 22 |
|---|---|
| Demographic and biological factors | |
| Health status | 12 (55%) |
| Medication side-effects | 5 (23%) |
| Psychological, cognitive and emotional factors | |
| Self-motivation | 10 (45%) |
| Mental illness stigma | 7 (32%) |
| Value of physical health benefits | 7 (32%) |
| Psychological health | 3 (14%) |
| Time availability | 2 (9%) |
| Cognitive function | 2 (9%) |
| Behavioural factors | |
| Physical activity enjoyment | 1 (5%) |
| Physical activity structure (e.g., intensity/duration) | 1 (5%) |
| Perceived suitability of activities | 1 (5%) |
| Social and cultural factors | |
| Social support from family, friends or peers | 9 (41%) |
| Acceptance of women exercising | 1 (5%) |
| Acceptance of group exercise | 1 (5%) |
| Physical environment factors | |
| Environment restrictions | 7 (32%) |
| Safe environment | 7 (32%) |
| Organisational factors | |
| Staff availability/capacity | 18 (82%) |
| Funding | 16 (73%) |
| Staff capability to support physical activity | 13 (59%) |
| Staff working collaboratively | 4 (18%) |
| Therapeutic relationship with staff | 4 (18%) |
| Links to community groups/services | 1 (5%) |

motivation was the second most highly cited correlate of physical activity overall. Similar findings were observed among the 22 qualitative studies that reported on correlates, with the addition of social support. There was also more emphasis on stigma, health, value of health and funding for physical activity programmes in the qualitative studies. Overall, a broad range of correlates have been reported.

One hundred and thirty-nine interventional studies were identified; only eight of these were larger-scale RCTs with 100 or more participants, and long-term follow-up was rare with only six out of the 57 RCTs (11%) having a follow up of 6 months or more. We did not identify any studies that included an economic evaluation, and there were few replications, suggesting that it would be challenging to evaluate the clinical and cost-effectiveness of physical activity interventions within the current evidence.

In terms of intervention design, the majority of studies explored an intervention that involved delivering physical activity (81%) rather than promoting physical activity (8%); 9% used a mixed model of both delivering and promoting physical activity. Interventions were mainly limited to the period of inpatient stay. The type of physical activity delivered was generally some form of structured exercise or physical recreation targeting individuals. However, descriptions of the interventions were often incomplete, and interventions appeared to lack

**Table 5. Themes of research recommendations from studies of physical activity correlates.**

| Theme | Examples of specific research recommendations | Number (%) of articles out of 61 |
|---|---|---|
| **Research topic-related recommendations** | | |
| Investigate other groups | "it seems worthwhile to replicate this study in out-patients" [35] "it might be preferable to compare more extreme groups" [35] "subgroups of patients for example, with persons with severe depression" [36] | 6 (10%) |
| Investigate other correlates | "implicit attitudes towards exercise" [35] "consider other comparisons such as by body mass index, perceived physical health or age" [37] "the role of systemic influences on mental health treatment for exercise facilitation" [38] "investigation of non-linear relationships between the physical activity and the basic psychological needs is warranted" [39] | 6 (10%) |
| Investigate other setting | "consider staff perceptions across other healthcare settings" [40] | 2 (3%) |
| **Research methods-related recommendations** | | |
| Use other data collection methods | "incorporating objective measurements of physical activity and sedentary behaviour" [41] "A more structured questionnaire or short interviews with patients might have allowed for the collection of richer descriptions" [42] "More extensive interviews with the patients and clinical staff and longer period of participant observation" [43] | 7 (11%) |
| Larger studies | "with larger samples of patients" [36] "replicate the findings with larger sample sizes" [39] | 5 (8%) |
| Longitudinal studies | "the examination of cross-lagged relationships between these variables over time" [35] "further (longitudinal) research to obtain more accurate insights on this topic" [44] | 5 (8%) |
| Intervention studies | "explore the most effective exercise promotion strategies for autonomous exercise engagement for long lasting behaviour change" [40] "identify effective exercise interventions and feasible delivery models" [45] | 5 (8%) |
| Control for additional factors | "controlling more systematically for cognitive capacity" [35] "while controlling for the effects of medication and symptom severity" [41] | 2 (3%) |
| Explore mechanisms | "regarding the mechanisms to modify such views as a means of increasing such care provision" [46] | 1 (2%) |
| **Patient involvement-related recommendations** | | |
| Intervention co-produced with patients | "Patient led ideas and co-design should be key principles in programme and environmental design" [47] | 1 (2%) |

systematic development. For example, within the 57 RCTs, only two interventions were described as being based on a specific theory [70,71], and only one [72]made reference to the Medical Research Council's guidance on developing and evaluating complex interventions [73]. There was a notable lack of research on environmental interventions and interventions targeting healthcare professionals. In the current review, and elsewhere [74], we have observed that the reporting of interventions is often inadequate. Since adequate reporting of interventions is central to interpreting study findings and translating effective interventions into practice, we recommend that researchers use relevant reporting guidelines, such as TIDieR [75], when writing study reports.

Research has been conducted in a wide range of countries, including both high-income and low- and middle-income countries. However, the majority of studies were conducted in

**Table 6. Themes of findings arising from qualitative studies of physical activity interventions.**

| Theme | Examples of specific findings | Number (%) of articles out of 14 |
|---|---|---|
| **Experiences of the intervention** | | |
| Feasibility and acceptability | "Service users gave positive feedback regarding the programme" [48] <br> "The intervention was feasible as part of treatment" [49] | 6 (43%) |
| Physical impacts | "Improvements in physical function" [50] | 5 (36%) |
| Psychosocial impacts | "participants experienced interconnectedness and developed a mindful stance" [51] <br> "Gaining a sense of competence" [50] <br> "more relaxed, happier and calmer after training sessions" [48] | 11 (79%) |
| **Factors influencing intervention delivery and participation** | | |
| Barriers to participation | "medication side-effects" [52] <br> "the locked environment" [52] <br> "ill health" [50] <br> "low confidence" [50] <br> "perceived not physically able to undertake some activities" [50] <br> "low motivation" [53] | 6 (43%) |
| Facilitators of participation | "Enjoyment and some autonomy for service users" [52] <br> "physical, mental and social benefits of participating" [52] <br> "staff and patients exercising together" [45] <br> "exercise being a mandatory part of treatment" [45] <br> "instructor who is trained" [45] | 6 (43%) |
| Factors impeding implementation/delivery | "lack of funding" [52] <br> "staff resistance due to exercise apathy in their own lives" [52] <br> "lack of physical space and facilities" [52] | 2 (14%) |

European psychiatric inpatient settings. Given the fact that forensic settings are likely to have a more stable population, in terms of patients' length of stay, than some other inpatient settings, where patients may stay only a few days, surprisingly few RCTs have been conducted in forensic settings (n = 1).

Outcomes collected were mainly mental or physical health-related and the primary outcome was often not clearly stated. In many studies more than one mental or physical health outcome was examined, with some of the outcomes being positive and others neutral or negative, leading to difficulty interpreting the study results. It is important that studies, especially RCTs, clearly state their primary outcome from the outset to avoid cherry-picking results. Physical activity and health-related quality of life were rarely included as outcomes; the lack of physical activity outcomes suggests that increasing levels of physical activity was not the primary objective of many of the studies.

This current review differed in scope from the earlier papers [14,20,21] in that it encompassed varied settings (all adult inpatient mental health settings) and clinical populations (not only patients with serious mental illness but also those who receive care on inpatient mental health wards, e.g., those with dementia). In this way, it offered a systematically developed map of research available on physical activity interventions in adult users of inpatient mental health services. It also differed in its purpose and methods from previous studies [14,20]. For example, it included qualitative studies, which facilitated the identification of contextual factors that may influence the implementation and delivery of physical activity interventions in adult inpatient mental health settings.

## Strengths and limitations

This review encompasses a wide range of articles and was conducted rigorously and systematically following a predefined protocol and existing guidance for scoping reviews. It provides a

**Table 7. Themes of research recommendations from studies of physical activity interventions.**

| Theme | Examples of specific research recommendations | Number (%) of articles out of 139 |
|---|---|---|
| **Research topic-related recommendations** | | |
| Investigate other interventions/ intervention components | "Future interventions could use motivational interviewing" [53]<br>"the effect of different exercise modes" [54]<br>"explore other ways to foster regular physical exercise" [55] | 16 (11.5%) |
| Investigate other groups | "replicated in cohorts with affective disorders" [56]<br>"Further studies needed in women" [57] | 9 (6.5%) |
| Investigate mechanisms of effect | "consider the impact of symptom severity as a moderator" [56]<br>"elucidate other ways mindful-yoga impacts patient well-being and treatment" [58]<br>"explore the mechanisms of improved functional outcomes" [59] | 6 (4%) |
| Investigate optimal exercise strategy | "better identify specific exercise types, sets, repetitions, routines" [60]<br>"clarifying the type and dose of physical activity" [61]<br>"determining the ideal window for exercise intervention" [62] | 6 (4%) |
| Investigate other setting | "it might be better to explore music and exercise interventions in separate study settings to ease the realization of the interventions in a routine clinical setting" [63] | 1 (1%) |
| **Research methods-related recommendations** | | |
| Longer follow-up | "effect of inpatient self-selected exercise on affect and exercise adherence following discharge" [56]<br>"a longer follow-up period is needed" [58] | 17 (12%) |
| Investigate other outcomes | "consider including measures of affect and arousal" [56]<br>"other domains like cognitive performance, self-esteem, quality of life, stress and body image" [64]<br>"variables such as physiological measures should be included" [59] | 16 (11.5%) |
| Larger sample sizes | "Further studies in larger sample sizes are warranted" [65]<br>"a larger sample size is required" [59] | 15 (11%) |
| Use other data collection methods | "using objective measures of exercise intensity and sleep quality" [56]<br>"include more comprehensive quantitative measures" [58]<br>"using an extensive neuropsychological battery" [59] | 10 (7%) |
| Another study including a control group | "using a quiet rest group or an alternate activity group as a comparator" [56]<br>"a controlled research design" [58] | 8 (6%) |
| Replications/ confirmatory studies | "More studies to confirm the findings" [66]<br>"should be replicated in further clinical trial settings" [67] | 8 (6%) |
| Control for confounding variables | "Future studies would need to control for these variables as part of their analysis" [68] | 3 (2%) |
| Use a 'sham exercise' control group | "Future study designs might include fake conditions, such as . . . stretching and gymnastic sessions which do not boost the cardiorespiratory system" [69] | 1 (1%) |

comprehensive overview of the extent and type of research on physical activity in inpatient mental health services and identifies important gaps in the literature.

Although a systematic search was conducted, some eligible articles might have been missed. The review is also limited to English language articles and does not include grey literature. Although a scoping review was appropriate to meet the objectives of this study, it is subject to limitations that are typical of this approach. For example, it did not permit a quality assessment of the included studies. It was also limited in its capacity to conclude what factors affect participation in physical activity among adult users of inpatient mental health services, what interventions are effective, and what themes recur in the qualitative literature.

## Recommendations for future research

A 2018 meta-review of the evidence on physical activity as a treatment for SMI made several research recommendations [9], the following of which are relevant for research in relation to the inpatient context:

- More research is needed to establish pragmatic, scalable methods for delivering physical activity with the ultimate goal of optimising treatment 'reach'.

- More research is required to ascertain the optimal frequency, intensity, time and type of physical activity interventions in each inpatient sub-population, acknowledging that these might vary according to individual needs, preferences and characteristics.

- More research is needed to explore how specific individual, interpersonal, environmental, and organisational factors influence the uptake and maintenance of physical activity interventions when they are implemented in clinical practice.

- More research is needed to investigate whether sedentary behaviours can be reduced in inpatients and tease out the importance of reducing sedentary behaviours in structured exercise interventions.

- Economic evaluations are required to establish the cost-effectiveness of specific physical activity interventions.

In addition, we believe that more research is needed to develop and test interventions that span the transition from inpatient to community settings. For example, it would be useful to study longer-term outcomes, such as the impact of physical activity programmes on rates of readmission to inpatient mental health facilities. Finally, this scoping review has identified sets of articles that could be collated for quantitative or qualitative systematic reviews of physical activity in inpatient mental health services. Researchers conducting such reviews should consider the quality of the studies, which was beyond the scope of this review, and to consider the variety of populations represented in these studies.

## Conclusions

In conclusion, this scoping review summarised the extent and type of research on physical activity in adult users of inpatient mental health services. We identified a modest volume of evidence regarding correlates of physical activity. A broad range of individual, interpersonal, environmental and organisational correlates have been reported, but more studies with a longitudinal design are needed to determine how these and other factors are associated. We also found a large number of studies reporting on the outcomes of specific physical activity interventions. Most interventions targeted service users and involved the delivery of physical activity sessions rather than self-management interventions. There were very few large-scale RCTs and most studies did not include physical activity or quality of life outcomes. The findings of this review will help guide further primary research that is needed to guide clinical practice and policy.

## Supporting information

**S1 File. Search strategy.**
(DOCX)

**S2 File. Data extraction form.**
(DOCX)

**S3 File. Reference list of included reports.**
(DOCX)

## Author Contributions

**Conceptualization:** Garry A. Tew, Emily Peckham, Suzy Ker, Jo Smith, Philip Hodgson.

**Data curation:** Garry A. Tew, Emily Peckham, Suzy Ker, Jo Smith, Philip Hodgson, Katarzyna K. Machaczek, Matthew Faires.

**Formal analysis:** Garry A. Tew, Emily Peckham.

**Funding acquisition:** Garry A. Tew, Emily Peckham, Suzy Ker, Jo Smith, Philip Hodgson.

**Investigation:** Garry A. Tew, Emily Peckham, Suzy Ker, Jo Smith, Philip Hodgson, Katarzyna K. Machaczek, Matthew Faires.

**Methodology:** Garry A. Tew, Emily Peckham, Jo Smith.

**Project administration:** Garry A. Tew.

**Resources:** Garry A. Tew.

**Software:** Garry A. Tew, Emily Peckham.

**Supervision:** Garry A. Tew, Emily Peckham.

**Validation:** Garry A. Tew.

**Visualization:** Garry A. Tew.

**Writing – original draft:** Garry A. Tew, Emily Peckham, Katarzyna K. Machaczek.

**Writing – review & editing:** Garry A. Tew, Emily Peckham, Suzy Ker, Jo Smith, Philip Hodgson, Katarzyna K. Machaczek, Matthew Faires.

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
