## [Decision Letter · Decision Letter 0]

19 Jan 2024

PONE-D-23-36017Physical activity in adult users of inpatient mental health services: a scoping reviewPLOS ONE

Dear Dr. Tew,

Thank you for submitting your manuscript to PLOS ONE. After careful consideration, we feel that it has merit but does not fully meet PLOS ONE’s publication criteria as it currently stands. Therefore, we invite you to submit a revised version of the manuscript that addresses the points raised during the review process.

We look forward to receiving your revised manuscript.

Kind regards,

Maher Abdelraheim Titi

Academic Editor

PLOS ONE

Journal Requirements:

"This study was supported by Research Capability Funding from Tees Esk and Wear

Valleys NHS Trust"

Reviewers' comments:

Reviewer's Responses to Questions

**Comments to the Author**

1. Is the manuscript technically sound, and do the data support the conclusions?

Reviewer #1: Partly

Reviewer #2: Partly

2. Has the statistical analysis been performed appropriately and rigorously? 

Reviewer #1: Yes

Reviewer #2: N/A

3. Have the authors made all data underlying the findings in their manuscript fully available?

Reviewer #1: Yes

Reviewer #2: Yes

4. Is the manuscript presented in an intelligible fashion and written in standard English?

Reviewer #1: Yes

Reviewer #2: Yes

5. Review Comments to the Author

Reviewer #1: This scoping review has systematically collated qualitative and quantitative studies related to physical activity among adult users of mental health services within the context of inpatient settings. I believe the search strategy and processes for identifying articles are clear, transparent and replicable. This would be useful for researchers interested in accessing relevant literature and conducting future systematic quantitative and qualitative reviews in this area. The review has also highlighted a few important gaps and noted areas for future research. I believe the review has merit. However, I have outlined several comments that I believe could help to improve the quality of the manuscript that the authors may wish to consider.

A couple general comments:

1) Throughout the manuscript the term ‘correlates’ is used. I notice that in your protocol you refer to ‘determinants’. Does the approach taken here differentiate ‘correlates’ from ‘determinants’? My understanding is that some authors have used the terms ‘correlates’, ‘determinants’, ‘barriers’ and ‘facilitators’ differently to refer to specific types of findings.

It seems here that correlates is being used to refer to correlations and predictors from cross sectional research, predictors from longitudinal studies, factors identified in qualitative research (as identified by participant and/or authors?) (also, how is this different from the 'barriers' and 'facilitators' identified from the qualitative intervention findings)? Is that correct? Does it also include factors within intervention studies that authors have targeted or suggested as influencing PA? For example, if authors discuss targeting psychological needs as part of their intervention (e.g., autonomy, competence, relatedness), would this have been coded as a ‘correlate? I think it would be helpful to clarify to readers how you conceptualize ‘correlate’ in the manuscript.

2) You note that you extracted key findings relevant to the research questions (line 152). I was a bit unclear about what the questions were specifically when reading the intro and methods. In the discussion you had mentioned long term follow up was rare, and I was wondering if that was something that was extracted or just something that was found along the way post hoc. I checked and noticed that you did specify these in the protocol (what are the characteristics and findings of studies of PA interventions for this population? What PA support is offered to people after they have been discharged from the inpatient setting? What are the determinants of PA in adult users of inpatient mental health services?). It might be worth explicitly citing these questions to help clarify for readers the specific questions guiding the review process (in the introduction or perhaps the methods).

3) I think providing the data extraction sheet would also be helpful in increasing transparency and replication. It would help readers understand what was systematically extracted. For instance, in the discussion, it is noted that environmental interventions were uncommon, interventions targeting health professionals were rare, and (as previously mentioned) that long term follow up was uncommon. I wasn’t clear if this was information that had been systematically extracted during the review process or not.

Introduction

4) Line 43 – stress disorders (instead of stress disorder)

5) Line 70 – focused instead of focussed

Methods

6) What does ‘descriptive analytical approach’ refer to here? The Levac et al. source cited here describes a need for greater clarity regarding the 'descriptive analytic' approach referenced by Arksey and O'Malley and they mention the use of content analysis. It seems that Campbell et al. describe using content analysis, which can vary considerably in terms of approach used. Could you provide a bit more detail on the ‘descriptive analytical’ approach used in this review. Particularly, how were codes/categories identified? Were categories and codes derived inductively through familiarization with the data? Was this done by independent reviewers? Was any sort of software used (NVivo)? For the qualitative studies how were ‘correlates’ and themes identified?

Results

7) In table 2 and elsewhere (e.g., discussion lines 276-278), type of programme compared ‘PA delivered’ and ‘PA promoted’. To my mind PA promotion is a broad conceptualization including any efforts to encourage PA, and others have considered this to include PA counselling, prescription, referral and coaching/supervision. What is the distinction between these two in this review? Based on the context of this review it sort of seems like it might be referring to supervised (i.e., PA overseen by professional) vs unsupervised (i.e., counselling/advice/prescription administered by professional and then the person does the program on their own), but I’m not sure. Please clarify.

8) In table 2 under types of physical activities ‘various activities’ represents a fair share of the findings, but I was unclear about what this refers to. Does it mean a mix of the other activities described (e.g. yoga, sport, structured exercise)? Would it be possible to provide a brief description of the category in brackets (mix of other categories)?

9) Line 188 and tables 1 & 2. Also, in the discussion (lines 271-272): “only eight of these were large scale RCTs with 100 participants or more…”

Why were the cutoffs for sample size chosen? What is the relevance of a sample size of 100? Is a quality assumption being made here based on a sample size of less than 100? Was a decision made based on a statistical decision for research in this area? (example, following Cohen’s recommendations or some other guidelines?)

10) Lines 233-234- it is noted that “The duration, frequency, and intensity of session varied (see Table 2)”. I could not find information related to the frequency or intensity of activity interventions. Was this not mentioned in any of the studies or is it missing from the tables?

11) Lines 234-235 – Authors report that most interventions were delivered by health and/or exercise professionals. I could not find this information in the tables. In the tables, I see ‘qualified therapist’. Does this refer to mental health therapists, physiotherapists, or both? Could you provide the breakdown for exercise professionals and health professionals specifically in brackets?

12) Lines 235-236 – please provide percentages to support these findings (% of interventions that provided evidence on environmental interventions and interventions that provided participants with PA support following discharge).

13) I was uncertain about how some of these correlates were coded in relation to their categories. For example, I would have coded staff attitudes or professional capabilities (assuming you mean knowledge and skills?) as individual level factors, not organizational ones. Education and training, however, I would have coded as an organizational or system level correlate. I am surprised not to see education or training of professionals as correlates of PA, as previous systematic reviews have shown credentials/education of exercise specialists to be relevant in PA adherence among clients living with mental illness. Were these not identified in the included studies? Or did findings end up being coded as capabilities.? To my mind education/training are distinct from capabilities and others would agree (see Michie et al.’s Theoretical Domains Framework, for instance).

Also, I was unclear where the categories came from (i.e., demographic and biological, behavioural, etc.). Were they develop inductively based on findings or where they defined beforehand by a specific socioecological framework. For instance, I see no interpersonal factors domain (which is where I might have coded ‘therapeutic relationship’, which was coded under organizational factors in this review).

This gets at a broader point – were there any disagreements or challenges that faced authors they went through and identified the correlates? A bit of discussion on this point might be worthwhile in the limitation section. I think it is worth pointing out that the identification of the correlate list was a qualitative effort and some correlates/barriers that have appeared in other reviews may have been missed (e.g. education and training). This relates back to the point I made related to the methods and better clarifying the ‘descriptive analytic’ approach used.

Discussion

14) Lines 261-262 – How was the claim ‘organizational and cognitive, psychological, and emotional factors were the top two cited domains’ reached? For instance, I notice that the category that includes the highest percentage for the ‘correlates’, apart from organizational factors, is the category ‘demographic and biological factors’ (55% for health status). Some numbers supporting these statements in brackets might help readers understand how these claims are being reached.

15) Lines 268-270 - I would be a bit cautious here. The purpose of this review is to describe the nature and extent of the literature. As you have thoughtfully pointed out in limitations section, conclusions cannot be drawn about how various factors (or domains) affect participation. The factors that appeared most commonly may be due to what authors chose to measure or the focus of their questions.

I would consider refraining from statements that suggest the importance of certain domains here. I worry that some readers might walk away under the impression that this review provides evidence that organizational factors and cognitive, psychological, and emotional factors are the most important factors. If you decide to comment on this aspect here, I would follow it up with a clear caveat that the findings of this review do not support any claims of the relative importance of correlates or their categories. To do that, I think you would need studies that directly examine and compare all correlates/domains in relation to PA.

16) Lines 271-272 – I suggest specifying the number of trials that used long term follow up and defining what long term follow up means for readers in brackets (x% of interventions had a follow up period of at least 6 months).

17) Line 282- How was the conclusion ‘interventions lacked systematic development reached’ based on the findings from the review? Was information gathered concerning whether or not theory was used? Was it that specific outcomes weren’t decided a priori? What factors constituted ‘systematic development’? I think a bit of elaboration (just a sentence or two listing the factors that support this determination) may be helpful to readers.

Conclusion

18) Line 348 – It says here that ‘most interventions targeted the individual’, however, the results show that most interventions were group interventions (50%). Please clarify.

Other comments:

19) In the abstract it reads ‘more high-quality trials are needed to advance the field’, however, scoping reviews do not provide a quality analysis of included articles. I’m not sure about making this claim.

20) The reference list in the supplementary file could use an editing pass for formatting.

21) If there is room: In the introduction a few systematic reviews that have previously examined PA in inpatient mental health settings are noted. I think readers would be interested to know what this review offers above and beyond those previous reviews. My understanding from the introduction is that key points of distinction for this review is that it includes qualitative studies as well as different inpatient settings. Did the inclusion of qualitative findings and different inpatient settings unearth discoveries not covered in the previous reviews? If there is space left, I think it would be worthwhile to compare and contrast the main findings briefly and to state what this review offers above previous review findings.

Reviewer #2: I would like to thank the authors for providing me with the opportunity to provide this review. This field of interest is undeniably important with there being a continued need to identify effective ways to engage inpatient populations in physical activity.

It is clear that a significant amount of effort and expertise has been invested in this work and the authors should be commended for conducting a review of this size. My main concern is that I am struggling to see what the findings uniquely contribute to the research literature beyond that of other similar reviews. Specifically, it is not clear as to how the inclusion of literature beyond that of intervention research has contributed to our knowledge/understanding. I feel as if the wide breadth of literature that is collated means that some of the nuances that may be relevant for research and practice are lost. I believe that this work needs to be re-evaluated to see whether some key, meaningful findings for research and practice can be drawn out and explore whether it is possible to more distinctly articulate how the inclusion of how varied literature types provides new insights or advances to our understanding.

I have made some general comments for each section. Where my comment relates to a specific sentence, I have added the line number.

Abstract

Overall, this was well-written and clearly presented. In some areas, some more specificity on methods/key findings could have improved the abstract.

- Line 23 - Could you be more specific about what is meant by ‘including studies of physical activity’.

- Line 38 – recommendations for future research are vague. Could you be more specific about what future research should target.

Introduction

The introduction provides a good overview of the literature and you provide a sound rationale for conducting this review.

- Line 76 – I wonder if you could elaborate more on why the previous studies haven’t been able to identify the state of the literature/advance future research.

Methods

Methods are clearly described. I wonder if it is worth providing some overview on what is meant by inpatient settings. For example, perhaps providing a brief description of the varying levels of security.

Line 133: you only included studies that were post-2007. I appreciate it is challenging to select a date range that can capture all study types. However, As the mental health act relates to UK legislature, it might be worth highlighting that this date range would not apply to UK studies

Results

- The results section is comprehensively described. The tables are useful to present the large amount of information

- Line 203 onwards: You mention correlates of physical activity but don’t include much detail on how these correlates impact physical activity. This information would be really useful to help us understand areas of focus for future intervention.

Discussion

- There are some interesting findings that are well-described.

- Your recommendations for future research are not that strongly developed. It would have been more insightful to use the key findings from your paper to build these recommendations, rather that relying on recommendations that have already been created.

- You mention in your introduction (line 79) the benefits of including both qualitative and quantitative research. I would like to see this referred to here in the discussion– particularly what has been gained by including both research types.

6. PLOS authors have the option to publish the peer review history of their article (what does this mean?). If published, this will include your full peer review and any attached files.

Reviewer #1: No

Reviewer #2: No

---

## [Decision Letter · Decision Letter 1]

24 Mar 2024

Physical activity in adult users of inpatient mental health services: a scoping review

PONE-D-23-36017R1

Dear Dr. Tew,

We’re pleased to inform you that your manuscript has been judged scientifically suitable for publication and will be formally accepted for publication once it meets all outstanding technical requirements.

Kind regards,

Maher Abdelraheim Titi

Academic Editor

PLOS ONE

Reviewers' comments:

Reviewer's Responses to Questions

**Comments to the Author**

1. If the authors have adequately addressed your comments raised in a previous round of review and you feel that this manuscript is now acceptable for publication, you may indicate that here to bypass the “Comments to the Author” section, enter your conflict of interest statement in the “Confidential to Editor” section, and submit your "Accept" recommendation.

Reviewer #1: All comments have been addressed

Reviewer #2: All comments have been addressed

2. Is the manuscript technically sound, and do the data support the conclusions?

Reviewer #1: Yes

Reviewer #2: Yes

3. Has the statistical analysis been performed appropriately and rigorously? 

Reviewer #1: Yes

Reviewer #2: N/A

4. Have the authors made all data underlying the findings in their manuscript fully available?

Reviewer #1: Yes

Reviewer #2: Yes

5. Is the manuscript presented in an intelligible fashion and written in standard English?

Reviewer #1: Yes

Reviewer #2: Yes

6. Review Comments to the Author

Reviewer #1: Thank you for your revisions. I am satisfied with the changes made and believe this to be a worthwhile contribution to the literature.

Reviewer #2: I am satisfied that the suggestions have been addressed by the authors and recommend that this paper is accepted.

7. PLOS authors have the option to publish the peer review history of their article (what does this mean?). If published, this will include your full peer review and any attached files.

Reviewer #1: No

Reviewer #2: No

---

## [Editor Report · Acceptance letter]

26 Apr 2024

PONE-D-23-36017R1 

PLOS ONE

Dear Dr. Tew, 

I'm pleased to inform you that your manuscript has been deemed suitable for publication in PLOS ONE. Congratulations! Your manuscript is now being handed over to our production team.

Kind regards, 

on behalf of

Dr. Maher Abdelraheim Titi 

Academic Editor

PLOS ONE